# Coupling an Electroactive *Pseudomonas putida* KT2440 with Bioelectrochemical Rhamnolipid Production

**DOI:** 10.3390/microorganisms8121959

**Published:** 2020-12-10

**Authors:** Theresia D. Askitosari, Carola Berger, Till Tiso, Falk Harnisch, Lars M. Blank, Miriam A. Rosenbaum

**Affiliations:** 1Institute of Applied Microbiology—iAMB, Aachen Biology and Biotechnology—ABBt, RWTH Aachen University, 52074 Aachen, Germany; desy_askito@staff.ubaya.ac.id (T.D.A.); till.tiso@rwth-aachen.de (T.T.); lars.blank@rwth-aachen.de (L.M.B.); 2Leibniz Institute for Natural Product Research and Infection Biology—Hans Knöll Institute, 07745 Jena, Germany; carola.berger@leibniz-hki.de; 3Department of Environmental Microbiology, Helmholtz Centre for Environmental Research—UFZ, 04318 Leipzig, Germany; falk.harnisch@ufz.de; 4Faculty of Biological Sciences, Friedrich Schiller University Jena, 07745 Jena, Germany

**Keywords:** *Pseudomonas putida*, rhamnolipid, bioelectrochemical system, phenazines, redox mediator, microbial electrosynthesis, electrobiotechnology, metabolic engineering

## Abstract

Sufficient supply of oxygen is a major bottleneck in industrial biotechnological synthesis. One example is the heterologous production of rhamnolipids using *Pseudomonas putida* KT2440. Typically, the synthesis is accompanied by strong foam formation in the reactor vessel hampering the process. It is caused by the extensive bubbling needed to sustain the high respirative oxygen demand in the presence of the produced surfactants. One way to reduce the oxygen requirement is to enable the cells to use the anode of a bioelectrochemical system (BES) as an alternative sink for their metabolically derived electrons. We here used a *P. putida* KT2440 strain that interacts with the anode using mediated extracellular electron transfer via intrinsically produced phenazines, to perform heterologous rhamnolipid production under oxygen limitation. The strain *P. putida* RL-PCA successfully produced 30.4 ± 4.7 mg/L mono-rhamnolipids together with 11.2 ± 0.8 mg/L of phenazine-1-carboxylic acid (PCA) in 500-mL benchtop BES reactors and 30.5 ± 0.5 mg/L rhamnolipids accompanied by 25.7 ± 8.0 mg/L PCA in electrode containing standard 1-L bioreactors. Hence, this study marks a first proof of concept to produce glycolipid surfactants in oxygen-limited BES with an industrially relevant strain.

## 1. Introduction

Rhamnolipids (RL) are among the best explored microbial glycolipid surfactants. The combination of one or two hydrophilic rhamnose monosaccharides with lipid chains of different length and different degree of saturation (called congeners) [1,2], provides a large range of specific properties [3]. Rhamnolipids possess eco-friendly characteristics such as biodegradability, a low eco-toxicity, and they can be produced from renewable resources [4]. The expansion of their production to industrial scale is therefore desirable. However, the biotechnological production with the native microbial producers, especially with *Pseudomonas aeruginosa*, is not desirable. *P. aeruginosa* is a facultative pathogen and its rhamnolipid production is controlled by a complex quorum sensing network [5,6,7]. Hence, research in the past years has focused on heterologous production with the non-pathogenic *P. putida* KT2440 [8,9,10].

Metabolic engineering of *P. putida* KT2440 for rhamnolipid production was previously developed and tested under aerobic conditions [3,10,11] (Figure 1). However, the costly aeration and the subsequent problems with strong reactor foaming still impose severe technical challenges [12]. This also hinders the scale-up of the process towards the industrial production of rhamnolipids and it requires substantial technical effort to overcome this challenge [13,14,15,16]. To reduce this bottleneck, which comes with the need for sufficient oxygen supply for aerobic growth and product synthesis, we here explore an alternative approach for rhamnolipid production using *P. putida* as a biocatalyst under oxygen-limitation in bioelectrochemical systems (BESs). BESs couple the metabolic activity and product formation of microorganisms to an electron transfer with an extracellular electrode, for instance, the discharge of surplus metabolic electrons to an anode [17].

The implementation of the BES technology for rhamnolipid production is based on the successful heterologous production of phenazines serving as redox mediators in *P. putida* for mediated extracellular electron transfer (MET; Figure 1) [18,19]. Using MET, the obligate aerobic organism *P. putida* KT2440 was enabled to stay metabolically active under oxygen-limited conditions in a BES through the electron discharge to an extracellular anode. Metabolic activity was evaluated under different aeration scenarios, and it was shown that the best regime for phenazine- and current production was an initial active aeration of the media for 48 h, followed by passive aeration of the headspace through open vent filters until the end of cultivation. With this set-up, depending on the strain, the redox mediators phenazine-1-carboxylic acid (PCA) and/or pyocyanin (PYO) were continuously produced, and anodic current could be sustained for more than 10 days [18]. While this route of anodic MET in *P. putida* still is limited and work is ongoing to enhance the efficiency of electron discharge [19], the technical exploitation of the sustained metabolic activity can already be evaluated with a relevant biotechnological synthesis.

We here evaluated if both, phenazine mediator production and rhamnolipid production, can be heterologously engineered into a single strain of *P. putida* KT2440. The strain was first evaluated in classical fully aerobic cultivations and then further assessed for bioelectrochemical production with active aeration for 48 h followed by passive aeration (AA), a continuous active aeration (AA+), and passive aeration from the start (PA) in BES benchtop reactors. The most successful operational scheme was finally up-scaled and reconfirmed in 1-L electrobioreactors, which are continuous stirred tank bioreactors fully standardized for electrochemical operation [20]. The successful prove of concept portrait in this work, opens the door for a wider use of microbial electrosynthesis by decreasing the need for oxygen in otherwise obligate aerobic processes.

## 2. Materials and Methods

### 2.1. Bacterial Strains, Strain Cultivation and Media Preparation

The *Pseudomonas putida* strain KT2440 (DSM 6125) served as the recipient for the rhamnolipid biosynthesis genes *rhlAB* of *P. aeruginosa* PAO1 (DSMZ 19880) and the phenazine biosynthesis genes *phzA2-G2* of *P. aeruginosa* PA14 (DSMZ 19882, [19]). 250 mL shake flasks were used for standard strain cultivations with LB medium (Carl Roth, Karlsruhe, Germany) with or without the addition of antibiotics (as required). *P. putida* cultures were incubated at 30 °C and *P. aeruginosa* cultures at 37 °C. For all reported experiments (shake flasks, BES experiments and well plates), *P. putida* was grown at 30 °C in Delft mineral salt medium [21], in short, the final composition (per L) consisted of 3.88 g K_2_HPO_4_ (22 mM), 1.63 g NaH_2_PO_4_ (14 mM), 2 g (NH_4_)_2_SO_4_, 0.1 g MgCl_2_·6H_2_O, 10 mg EDTA, 2 mg ZnSO_4_·7H_2_O, 1 mg CaCl_2_·2H_2_O, 5 mg FeSO_4_·7H_2_O, 0.2 mg Na_2_MoO_4_·2H_2_O, 0.2 mg CuSO_4_·5H_2_O, 0.4 mg CoCl_2_·6H_2_O, 1 mg MnCl_2_·2H_2_O, with 5 g/L (27 mM) (passive aeration), 10 g/L (55 mM) (active aeration), or 5 g/L (27 mM) (electrobioreactors) glucose. Glucose concentration was lower at PA configuration, as the metabolic activity of *P. putida* is known to be greatly reduced at this condition [19]. Antibiotics were added as required (50 μg/mL kanamycin or/and 30 μg/mL gentamycin). In the electrobioreactors, 37.5 instead of 30 μg/mL gentamycin were used to increase plasmid stability.

### 2.2. Combining Heterologous Rhamnolipid and Phenazine Production in P. putida

All plasmids and strains used in this study are shown in Table 1. Phenazine production plasmids have been generated previously [19]. The genes *rhlA* and *rhlB* for rhamnolipid production originated from *P. aeruginosa* PAO1 and were previously cloned into the pSK02 plasmid [16], which served as the template for this study. Plasmids were assembled via Gibson assembly (New England Biolabs-Gibson Assembly) [22]. The *rhlAB* genes were inserted into the pJNN backbone and the pJNN.*phzMS* plasmid to generate pJNN.*rhlAB* and pJNN.*rhlAB.phzMS*, respectively (Appendix A). Subsequently, these plasmids were transformed into *P. putida* KT2440 via electroporation [23]. Transformed cells were selected on LB agar plates with gentamycin.

To verify the constructs, colony PCR, as well as restriction digest analysis, and DNA sequencing (GATC-Eurofins Genomics) was used. The respective primers for DNA sequencing are shown in Appendix A. Subsequently, the pBNT14.*phz2* plasmid was transformed into the *P. putida* RL and *P. putida* RL-MS strains via electroporation [23]. The selection procedure was identical to the *rhlAB* containing plasmids but using a combination of kanamycin and gentamycin for selection (50 μg/mL kanamycin and 30 μg/mL gentamycin).

### 2.3. Aerobic Strain Characterization and Evaluation

In order to evaluate the production of phenazines and/or rhamnolipids, a total of eight clones per constructed strain were analyzed in triplicates using a multiplexed micro-cultivation platform (CR1424d, EnzyScreen, Heemstede, The Netherlands), in 24 square-well plates filled with 5 mL Delft medium, with 20 mM added glucose, and supplemented with the respective antibiotics. Afterwards, 250 mL Erlenmeyer flasks; filled with 25 mL Delft medium were used for aerobic shake flask experiments. For both types of cultivations, the starting OD_600_ was 0.1. After three hours of incubation, 0.1 mM (for phenazine production) and 1 mM (for combined phenazine- and rhamnolipid production) sodium salicylate was added as an inducer for gene expression. The incubation temperature was always at 30 °C, while the shaking frequency depended on the type of cultivation and was 224 rpm for the multiplexed cultivation and 200 rpm for the shake flasks. The best performing clone of each constructed strain from the multiplex cultivation was selected and further characterized in aerobic shake flasks—as well as BES experiments.

### 2.4. Bioelectrochemical Experiments

The benchtop single chamber glass BES reactors (500 mL working volume) included a water jacket for temperature control [19]. A three-electrode set-up was used, including (i) a graphite comb of high grade graphite (EDM-3, Novotec) as the working electrode (anode, surface area: 156.32 cm^2^); (ii) a graphite block (made of the same material as the working electrode) as the counter electrode (cathode, surface area: 49.22 cm^2^); and (iii) a self-assembled Ag/AgCl, saturated KCl reference electrode (RE; 197 mV vs. SHE at 25 °C, all potentials are given vs. this reference). BES experiments were generally run with the anode poised at 0.2 V vs. RE. The RE was made by electrochemically oxidizing a silver wire at a potential >0.5 V for 10 min in a concentrated KCl solution. The wire was thereafter embedded into a saturated KCl agar matrix (~3–4 M KCl; ~1.5% [*w*/*v*] of agar), acting as the electrolyte, within a thin glass shaft. The bottom end of the shaft was a MgO frit to ensure ionic exchange between the reference electrode and the reactor medium. Potentials were verified against a commercial Ag/AgCl reference electrode before and after BES experiments. The reactors were controlled by a potentiostat (VMP3, BioLogic) at 0.2 V. To provide the best conditions for the bioelectrochemical production of rhamnolipids, several different strategies for oxygen limitation of the benchtop BES reactors were tested in triplicates: (i) active aeration via a sparger at a flow rate of 30 mL/min for 48 h, followed by passive aeration of the head space via two open vent filters (AA); (ii) continuous active aeration at a flow rate of 30 mL/min (or 50 mL/min where indicated) (AA+); or (iii) passive aeration via open vent filters for the entire experiment (PA). Oxygen levels were measured discontinuously for evaluation purpose in the benchtop BES reactors via a VisiFerm DO Arc sensor (Hamilton). For this set-up passive aeration corresponds to a range of 50–80 dissolved oxygen/ppb, while in active aerated reactors the dissolved oxygen/ppb was at 1500–2500.

For synthesis at a 1 L-scale, electrobioreactor were employed. They are based on commercial bioreactor vessels (Infors Multifors, Infors AG, Bottmingen, Switzerland) equipped with an electrochemical upgrade kit [20,24,25]. In brief, the kit consists of a custom-made polyetheretherketon (PEEK) reactor lid including all connectors and containing an inner compartment with three side openings (total window area 21.48 cm^2^) to allow mixing throughout the reactor. The reactor was filled with 750 mL working volume. The two 25 cm long graphite rods (10 mm diameter, CP-2200 quality, CP-Handels GmbH, Wachtberg, Germany) acting as working electrodes were placed in the outer anode compartment (total surface area 76.97 cm^2^), while two identical rods were located within the inner cathode compartment. A commercial Ag/AgCl reference electrode was used (sat. KCl, SE11, with 6 mm diameter, and 100 mm length, Xylem Analytics, Meinsberg, Germany). Additionally, the pH, temperature and pO_2_ were continuously measured using the integrated sensors of the Infors Multifors system. Here, pO_2_ is measured via a Clark electrode. If the pH dropped below a value of 6.5, it was titrated with 10% NaOH solution. These reactors were also potentiostatically controlled (VMP3, BioLogic) at an anodically applied potential of 0.2 V vs. RE and operated with the AA or the PA set-up. As the pO2 sensors in the electrobioreactors measure the off gas, they need a constant airflow and could therefore record no values at PA set-up. For the initially active aerated electrobioreactors pO2 was down to 0.06% for the metabolically active phase.

Cultivation temperature for all BES experiments was 30 °C at a stirring speed of 200 rpm via a magnetic stirrer. The amount of produced current (chronoamperometry) was recorded continuously (once every minute), after 24 h of blank media measurement. Sampling for pH, HPLC analysis, and optical density measurements at 600 nm was performed daily. Except when indicated otherwise, all experiments were run as independent biological triplicates. Where applicable, the data is given with standard deviation.

### 2.5. Analysis of Sugar Metabolites

The amount of glucose and secreted metabolites was measured via an HPLC containing a 300 × 8.0 mm polystyroldivinylbenzol copolymer separation column (CS-Chromatographie, Langerwehe, Germany). For the detection, a UV/VIS detector at 210 nm and a refractive index (RI) detector at 35 °C were employed. The mobile phase consisted of 5 mM sulphuric acid applied at an isocratic flow rate of 0.8 mL/min, at a separation temperature of 60 °C. As standard solutions, analytical grade glucose, gluconate, 2-ketogluconate, and acetate (all Carl Roth) were used. Due to a change in the lab facility, the metabolites of the electrobioreactor experiments were measured on a Aminex HPX-87H ion exclusion column (300 mm × 7.8 mm, 9 µm; Bio-Rad, Hercules, CA, USA) and a Kromasil 100 C (18, 40 mm × 4 mm, 5 µm; Dr. Maisch GmbH, Ammerbuch-Entringen, Germany) precolumn. The separation occurred at a flow rate of 0.5 mL/min at 50 °C. Detection method and running buffer were identical.

### 2.6. Phenazine Analysis

Phenazines were quantified via an HPLC equipped with a C18 column (Waters Corporation) coupled with a photodiode array UV/VIS detector (LC-168, BecKann, Brea, CA, USA). Detection and quantification of PCA was achieved at 366 nm and of PYO at 280 nm. External standard solutions were prepared with analytical grade PCA (Princeton Biomolecular, Princeton, NJ, USA) and PYO (Cayman Chemical). For separation, a gradient of 0.1% trifluoroacetic acid (TFA) in acetonitrile (solution A) and 0.1% TFA in water (solution B) was used as the mobile phase (regime: 15% solution A for 2 min, followed by 100% for 15 min, and finished at 5% for 3 min) at a column temperature of 20 °C and a flow rate of 0.8 mL/min [26]. Due to a change in the lab facility, PCA in the electrobioreactor experiments was measured on a Prontosil 120-5-C18 ACE-EPS 250 × 4.0 mm (BISCHOFF Analysentechnik, Leonberg, Germany) column on a JASCO HPLC system equipped with a multi wavelength detector (MD-1515, JASCO International Co, Tokyo, Japan). The regime of solution A and B was changed to 10% solution A for 2 min, 50% for 10 min, 70% for 8 min, and 100% for 7 min at a flow rate of 1 mL/min.

### 2.7. Rhamnolipid Analysis

The quantification of rhamnolipids was performed using a reversed-phase HPLC equipped with a NUCLEODUR C18 Gravity column (Macherey-Nagel GmbH & Co. KG, Düren, Germany) (dimensions: 150 × 4.6 mm; particle size: 3 µm). The Ultimate 3000 HPLC system was equipped with a Corona-charged aerosol detector (CAD) (Thermo Fisher Scientific Inc., Waltham, MA, USA). The flow rate was 1 mL/min at 40 °C. Separation was achieved with a gradient of 100% acetonitrile (eluent A) and ultra-pure water containing 0.2% (*v*/*v*) formic acid (eluent B) as a buffer (70% eluent A for 9 min, 100% for 3 min, and 70% for 3 min) [10]. Due to a change in the lab facility, rhamnolipids of the electrobioreactor experiments were measured on a HPLCJAS2 system (JASCO International Co, Tokyo, Japan) with a NUCLEODUR C18 Gravity column (250 × 2 mm, 3 µm; Macherey-Nagel GmbH & Co. KG, Düren, Germany). Detection was achieved via an evaporative light scattering detector (ELSD) (Evaporator 80 °C, Nebulizer 30 °C, Gas flow 1 SLM) at 35 °C. The regime was changed to 50% eluent A for 25 min, a ramp to 100% for 2 min, running at 100% for 5 min and a 2 min ramp down to 50%.

## 3. Results

### 3.1. Tailoring Heterologous Mono-Rhamnolipid Synthesis with Phenazine Production in P. putida

Our work started with a PCA producing strain of *P. putida* KT2440 carrying the plasmid pBNT.14*phz2* for the expression of the genes *phzA2-G2* from *P. aeruginosa* PA14, as this strain showed superior phenazine production and bioelectrochemical activity in our previous work [19]. The genes *rhlAB,* responsible for mono-rhamnolipid expression in *P. aeruginosa* PAO1, were cloned into the salicylate inducible plasmid pJNN. Additionally, the previously used plasmid pJNN.*phzMS*, carrying genes required to turn PCA into PYO [18], was modified to contain *rhlAB* resulting in the pJNN.*rhlAB*.*phzMS* plasmid. These plasmids were separately transformed into the PCA producing *P. putida* KT2440 strain. The first resulting strain, *P. putida* RL-PCA, carrying pBNT.14*phz2* and pJNN.*rhlAB*, should be able to produce mono-rhamnolipids and PCA. The second strain, *P. putida* RL-PYO, containing pBNT.14*phz2* and pJNN.*rhlAB*.*phzMS,* should represent a mono-rhamnolipid, PCA, and PYO producer. As a control, both pJNN plasmids were also transformed into the *P. putida* KT4220 wildtype to evaluate rhamnolipid production without phenazines (see Table 1 for details).

The growth behavior and product formation of eight picked colonies was compared on a multiplexed micro-cultivation platform [27] under aerated conditions. While *P. putida* RL-PCA produced a maximum of 110.0 ± 0.5 mg/L mono-rhamnolipids and 54.4 ± 8.4 mg/L PCA, *P. putida* RL-PYO on the other hand only produced a maximum of 50.0 ± 10.0 mg/L mono-rhamnolipids, 48.8 ± 0.2 mg/L PCA, and 14.7 ± 0.3 mg/L PYO (Figure 2). In contrast, the best performing clone of the control strain *P. putida* RL produced 730 mg/L mono-rhamnolipids in LB media and 360 ± 30 mg/L in Delft minimal medium containing 10 g/L of glucose after 30 h of cultivation, respectively (Appendix A and Figure 2). All four common mono-rhamnolipid congeners naturally produced by *P. aeruginosa* were present in the engineered *P. putida* strain (i.e., Rha-C10-10 (predominantly), Rha-C10-C8, Rha-C10-C12, and Rha-C10-C12:1). For the *P. putida* RL-MS control strain, the best performing clone produced a maximum of 840 mg/L mono-rhamnolipids in LB media (Appendix A) and 400 ± 20 mg/L in the Delft minimal media (Figure 2), while some of the tested clones did not produce any mono-rhamnolipids. This shows the great variability of successful mono-rhamnolipid production.

The best performing *P. putida* RL-PCA and *P. putida* RL-PYO clones, as well as the non-phenazine controls from the micro-cultivation, were further characterized under aerobic conditions in shake flasks (Figure 3). Here, *P. putida* RL-PCA was able to produce a maximum of 141 mg/L mono-rhamnolipids and 57.7 ± 18.4 mg/L PCA after 48 h of fully aerobic cultivation. Thereby, the mono-rhamnolipid production of this strain was higher than that of the non-PCA producing *P. putida* RL strain, which only produced 110.0 ± 0.3 mg/L mono-rhamnolipids. By doubling the concentration of the inducer salicylate (to 2 mM), mono-rhamnolipid and PCA production further increased up to 152.0 ± 1.2 mg/L and 122.0 ± 12.5 mg/L, respectively. In the same set-up, the *P. putida* RL-PYO clone only produced 50 mg/L mono-rhamnolipids and 67.2 ± 7.8 mg/L PCA, as well as 34.0 ± 2.8 mg/L PYO. Here, the increase of the inducer concentration did not enhance the mono-rhamnolipid production, but increased PCA production to 130.8 ± 16.8 mg/L. Further, a reduced overall growth, as seen by the lower OD_600_ values, compared to the *P. putida* RL-PCA and *P. putida* RL strains was recorded. We therefore concluded that the ability of *P. putida* to produce PYO is of no advantage, but rather has a negative impact on RL production. Hence, the *P. putida* RL-PCA strain was chosen to be further characterized for bioelectrochemical production of rhamnolipids.

### 3.2. Bioelectrochemical Production of Rhamnolipids in Oxygen-Limited BES

#### 3.2.1. Applying Active Aeration to the Benchtop BES Reactors

We have previously shown that the highest PCA- as well as current production can be achieved when *P. putida* PCA is cultivated in an initially actively aerated BES, which is switched to passive aeration after 48 h (active aeration (AA) set-up) [19]. With this configuration an increased biomass formation is enabled during the active aeration period, as *P. putida* is an obligate aerobic organism, and the cells have a high metabolic activity when oxygen is available in sufficient amounts. This increased quantity of cells can than produce an ample concentration of PCA via plasmid-based expression. In order to mediate electrons to the anode and to generate a detectable current signal, rather than passing them onto oxygen, the concentration of the latter needs to be reduced. Hence, during the passive aeration phase, when metabolism is slowed, the anode is more incorporated, and the viability of the cells is secured for a longer time period. Therefore, the AA set-up was our initial configuration of choice for the bioelectrochemical characterization of the strains. As a control experiment, first the *P. putida* RL strain, solely able to produce mono-rhamnolipid but no PCA was characterized in AA-mode BES reactors at 0.2 V and without an applied potential to the electrode; hence, at open circuit potential (OCP; Appendix A). Here, no mono-rhamnolipids were produced. Next, the *P. putida* RL-PCA strain was tested in AA-mode BES reactors at OCP (Appendix A) and also did not produce any detectable amounts of mono-rhamnolipids under these oxygen-limited conditions. After these control experiments, we tested if the presence of a redox mediator and an anode poised at 0.2 V in an oxygen-limited BES would be beneficial for *P. putida* RL-PCA cultivation, as the electrode could potentially be used as an alternative terminal electron acceptor to support rhamnolipid production.

Therefore, we first tested the performance of the generated *P. putida* RL-PCA strain at 0.2 V with the AA regime (Appendix A). Here, the strain produced 75.7 ± 13.8 mg/L PCA, resulting in a current density of 10.6 ± 3.2 µA/cm^2^, which was in a similar range as in our previous work [19]. In 10 days, 89% of the provided glucose was consumed with minor amounts of 2-ketogluconate, an oxidation product of glucose [28], formed during the first three days. However, no detectable amount of mono-rhamnolipid was produced by the *P. putida* RL-PCA strain in this set-up during the 10 days of operation.

In the following, we switched the operation to 50 mL/min continuous active aeration throughout the entire experiment (AA+). This configuration aimed at providing higher amounts of oxygen to be available as electron acceptor for cellular catabolic reactions to support the production of mono-rhamnolipids (Appendix A). While the PCA production was increased by 16% compared to the AA regime, still no mono-rhamnolipids were produced. The increased amount of PCA led to a 30% increase in the maximum current density to 13.7 ± 1.8 µA/cm^2^. This was achieved with a glucose consumption of 93%. This result indicates that the lack of oxygen was likely not the main limiting factor for the production of mono-rhamnolipids in the BES reactors.

#### 3.2.2. Applying Passive Aeration to the Benchtop Single Chamber Glass BES Reactors

To complete our evaluation of different aeration regimes in BESs, we finally also operated the reactors with a continuous passive aeration (PA) of the headspace, i.e., providing even less oxygen to the culture than before. Surprisingly, under this stringent oxygen-limited condition, *P. putida* RL-PCA was able to produce low concentrations of mono-rhamnolipids of up to 30.4 ± 4.7 mg/L. It also produced 11.2 ± 0.8 mg/L of PCA, which resulted in a maximum current density of 8.1 ± 0.1 µA/cm^2^, at a substantially lower consumption of glucose, with 41% of the initially provided carbon source still present by the end of the experiment after 10 days (Figure 4a–c). Note that the reactors had a substantially lower cell density with an OD_600_ of ~1, as compared to the shake flasks experiments (Figure 3), partially accounting for the low rhamnolipid titer. This reduced OD_600_ is most likely due to an overall decreased biomass formation, as *P. putida* is known to be an obligate aerobic organism [29]. The dominating rhamnolipid congener still was Rha-C10-C10 with about two-thirds of the total mono-rhamnolipids, followed by Rha-C10-C12, RhaC10-C12:1 and only minute amounts of Rha-C10-C8. The experiment was repeated independently in triplicates with very similar results (Appendix A).

#### 3.2.3. Scaling-Up and Verification of Results in 1-L Electrobioreactors

To evaluate if a BES-based oxygen-limited rhamnolipid production can be robustly implemented in a more controlled setting, we applied the *P. putida* RL-PCA strain in 1-L electrobioreactors [20]. In the passively aerated benchtop reactors the pH decreased to 6.09 ± 0.7 by the end of the experiment, and low pH can be growth inhibiting [30]. In the electrobioreactors, the pH was controlled to not drop beneath 6.5 and the experiments were conducted over a longer period of time (Figure 4d–f). The electrobioreactors were inoculated with a higher cell density as compared to the benchtop reactors (0.4 vs. 0.1, respectively) and a higher final cell density was reached (1.7 vs. 1.2). PCA was produced in comparable amounts in the two set-ups. After 10 days, which was the length of the benchtop BES reactor experiment, 15.4 ± 1.0 mg/L PCA was produced in the electrobioreactor. This value increased to up to 25.7 ± 8.0 mg/L by the end of the experiment after 25 days. After 10 days, 51% of the available glucose was consumed in the passively aerated electrobioreactor, whereby more 2-ketogluconate was produced than in the benchtop reactors. By the end of the experiment, after 25 days, glucose was completely consumed, while some 2-ketogluconate remained unused (3.7 ± 0.5 mM). Most importantly, we were able to verify the production of rhamnolipids. They were mainly produced in the first 24 h of the experiment with up to 30.5 ± 0.5 mg/L, which is very similar to the benchtop titers. After this initial increase, the detectable amount of rhamnolipids gradually declined until the end of the experiment. While in one of the duplicate reactors still 45% of the originally produced mono-rhamnolipids could be detected, no rhamnolipids were measurable in the second reactor on day 25.

To verify that indeed rhamnolipid production was only possible when the electrobioreactors were operated at passive aeration, a control experiment with applying active aeration for the initial 24 h (controlled at 0.8 L/min) was conducted (Appendix A). Like in the benchtop BES (Appendix A), PCA was produced throughout the experiments, reaching titers of up to 100 mg/L, while no rhamnolipids could be detected.

## 4. Discussion

We here describe the production of mono-rhamnolipids in the industrially relevant strain *P. putida* KT2440 under stringent oxygen-limited conditions. To achieve this, we enabled an alternative electron discharge of surplus metabolic electrons via electron shuttling to an extracellular electrode of a BES with heterologously produced phenazine redox mediators. The successful introduction of the *rhlAB* operon, originating from *P. aeruginosa* PAO1, into *P. putida* KT2440 for the heterologous production of rhamnolipids has been described previously (Table 2). This non-pathogenic bacterium possesses both critical pathways for the rhamnolipid precursor biosynthesis, activated rhamnoses and hydroxyalkanoic acids (Figure 1), respectively and grows well on glucose [9,10,11]. It is therefore a more suitable host than *E. coli* for a straight forward biodetergent production within a bioeconomy strategy. The here developed strain *P. putida* RL-PCA is additionally equipped to produce the *P. aeruginosa* PA14-derived phenazine PCA (Figure 1). We also constructed a *P. putida* strain, which has the genetic capacity to produce PCA as well as PYO in addition to the rhamnolipids (*P. putida* RL-PYO). These strains were initially characterized in fully aerobic cultivations to evaluate the impact of a co-production of phenazines on rhamnolipid synthesis. The concentrations of rhamnolipids in complex LB medium were generally 2–3 times higher than in minimal Delft medium. However, for a more quantitative evaluation of the subsequent bioelectrochemical experiments, the Delft medium was chosen as the main working medium. The non-phenazine producing control strains synthesized between 110 and 140 mg/L mono-rhamnolipids in this medium in shake flasks (Figure 3). This value was slightly increased when combined with PCA production, but drastically decreased when also PYO was produced (to only 35% of the *P. putida* RL-PCA level). Of the two phenazines, PYO is known to be more reactive and induces higher levels of oxidative stress [31,32,33]. Typically, also lower concentrations of PYO than of PCA are synthesized and tolerated by the cells to adjust to the increasing toxicity of PYO at higher concentrations. This increased stress in the presence of PYO might explain the reduced rhamnolipid production in the *P. putida* RL-PYO strain. We therefore focused our further investigations on the strain producing PCA as the sole phenazine (*P. putida* RL-PCA). During all aerobic cultivations of our newly engineered *P. putida* strains, we found all four common mono-rhamnolipid congeners that are naturally produced by *P. aeruginosa* in the expected ratio (i.e., predominantly Rha-C10-10, followed by Rha-C10-C8, Rha-C10-C12, and Rha-C10-C12:1) [34].

The then following evaluation of rhamnolipid production under oxygen-limited BES conditions led to surprising outcomes. In previous studies, we had assessed the oxygen-limited growth and phenazine-based electron discharge under different oxygenation levels [18,19]. There, we found the best performance with a mixed regime of an initial low-level aeration of the medium for biomass growth followed by a passive aeration of the headspace for current production. Completely anaerobic growth and maintenance of *P. putida* cells is generally not possible, since many essential metabolic reactions in this organism depend on oxygen [29]. Therefore, we first reproduced this initial active aeration protocol and while both, growth and PCA production were similar to the previous results [19], no rhamnolipids were produced. Anticipating that the limitation of oxygen was the reason for the absence of rhamnolipid production, we increased the aeration to a continuous medium level (50 mL/min) throughout the experiment; however, this did not lead to rhamnolipid production. We finally switched to our most constraint aeration regime, where only passive aeration of the headspace provides some oxygen. Despite the expected slower growth, glucose consumption, and the therefore reduced PCA synthesis (to only about one third), mono-rhamnolipids were produced. The concentration was only about 15–20% compared to the aerobic cultivations in shake flasks, but a reduced production under energetically very difficult circumstances was expected. We scaled this experiment up to 1 L using fully integrated electrobioreactors, which combine the electrochemical control with classical state-of-the-art bioreactor operation, including pH and oxygen control. Our results from the benchtop BES were fully confirmed—no rhamnolipid production with active aeration and—almost identical—surfactant production with only passive aeration of the reactors. These experiments ran more than twice as long and for the stationary phase we observed a slow degradation of the produced rhamnolipids.

We hypothesize that during active aeration of the culture, oxygen still was an extremely efficient electron acceptor and although glucose metabolism and respiration processes slowed down compared to shaken cultures, there was no overload with reducing equivalents within the cell. In contrast under passive aeration, the availability of oxygen to the cells was purely determined by diffusion and mixing of the BES liquid volume and oxygen became very scarce. Reducing equivalents like NAD(P)H generated from glucose, had to find alternative pathways for regeneration. One of these sinks for NAD(P)H utilization is the synthesis of activated hydroxyalkanoic acids in lipid *de novo* synthesis, which in turn are substrates for rhamnolipid production. Thus, while a part of the reducing equivalents was regenerated by charging the phenazines that were subsequently discharged at the anode, more regeneration was enabled by storing the electrons away in rhamnolipids. We know from our own and independent other group’s work that naturally phenazine-based extracellular electron discharge in *Pseudomonas* is restricted. The coulombic efficiency (fraction of electrons from the substrate harvested at the anode) is typically <10% [18,19,35] (here, maximum ~4.3% for passive aeration) and physiological investigations showed that while cells can maintain their metabolic functions, only a low level of ATP production and no growth is possible [18,19,36,37]. Several efforts are on the way to determine the physiological interaction partners of phenazines and to clarify the native electron transfer route. Research also focuses to re-engineer this extracellular electron transfer to be more efficient for the cell. However, based on what we found here, this should be very carefully controlled. Opening the doors to extracellular electron transfer too much might lead to a similar result as in actively aerated BES, were the glucose substrate is fully respired and no reduced rhamnolipid by-products are produced. Thus, a medium level respirative energy generation for controlled restricted growth at a surplus of reducing equivalents, which push for detergent production, might be the best scenario for this specific metabolic product. Especially for passive aeration, the amount of oxygen available to the cell will greatly depend on external factors like the reactor geometry and the size of the vent filters. With the data measured in this study, it is not possible to appoint a specific set-point for the oxygen concentration needed for rhamnolipid production. During passive aeration in the benchtop BES, we are very close to the detection limit of the used oxygen sensors and in the electrobioreactors no measurement was available during PA set-up. However, a basal requirement for oxygen for many metabolic functions within *P. putida* is certain [29]. To our knowledge, no study further determined how high this minimum obligate O_2_ concentration needs to be in order for this organism to sustain a fully functional cellular metabolism. This important parameter needs to be determined in the future to make electrobioproduction with *P. putida* more applicable.

To relate the rhamnolipid production of our strains to other attempts of heterologous rhamnolipid synthesis, we compare the mono-rhamnolipid titers for productions with *Pseudomonas putida* strains (Table 2). With glucose, titers of other strains, which have been engineered and further optimized for rhamnolipid production, reach 1.25–3 g/L in complex LB media. Our plasmid-based synthesis of rhamnolipids (without phenazines) reached about 0.75 g/L in complex media. Thus, for complex media, we are close to the state-of-the-art of heterologous rhamnolipid synthesis without any further strain optimization. In minimal media, the reached titers were with up to 0.36 g/L (without phenazines) and 0.15 g/L (with phenazines) still quite a bit lower. However, all those reported data are still far below an economic range and more strain engineering as well as bioprocess development work is required. Compared to the aerobic synthesis of rhamnolipids in minimal medium, the strictly oxygen-limited production in BES only reached titers of 0.03 g/L, i.e., one fifth of the aerobic concentration. While this is much lower, it should be recognized that hardly any oxygen as energetic driver was required for this production.

To conclude, we here showed for the first time the production of a biotechnologically relevant product under strictly oxygen-limited conditions with an obligate aerobic host using microbial electrosynthesis. Cellular maintenance energy was generated through extracellular electron transfer to an anode, which allowed for surplus reducing equivalents to be put forward to product synthesis. Thus, this work opens doors to oxygen-limited (and maybe in future anaerobic) production of highly energetic products, which normally require a well energized cell metabolism. With further improvements of the energetic interaction between *Pseudomonas* cells and the anode as electron acceptor, this process can be further enhanced, and relevant product concentrations can likely be reached. Beyond this work, this study also opens the path to the electrobiotechnological production of other types of glycolipid surfactant under oxygen limitation; or entirely different products, which might suffer from the presence of oxygen, such as redox active metabolites or proteins.

## Figures and Tables

**Figure 1 microorganisms-08-01959-f001:**
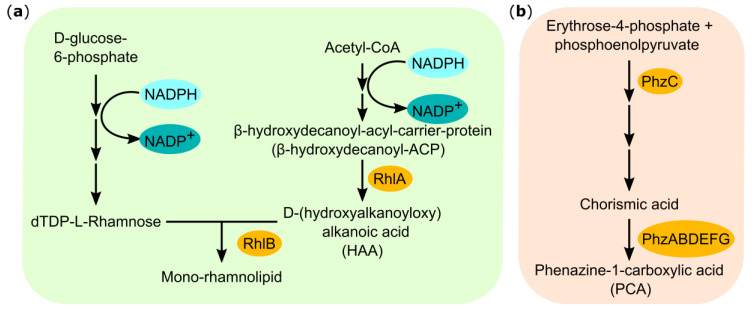
Schematic representation of the (**a**) rhamnolipid production and (**b**) phenazine biosynthesis pathways in *P. putida* KT 2440. The proteins needed for the heterologous production are highlighted in orange and were expressed via two plasmids.

**Figure 2 microorganisms-08-01959-f002:**
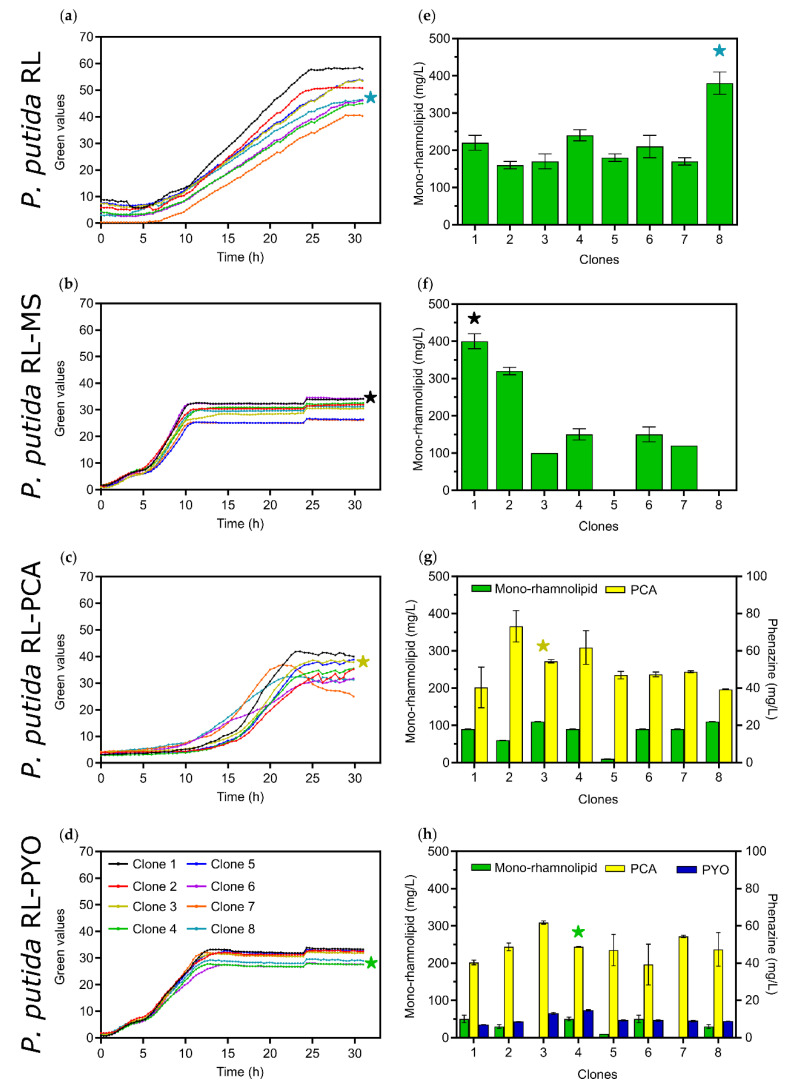
Growth behavior (**a**–**d**) and mono-rhamnolipid production (**e**–**h**) as well as phenazine production (**g**–**h**) of eight clones from each strain used in this study (*P. putida* RL, *P. putida* RL-MS, *P. putida* RL-PCA, *P. putida* RL-PYO; for detailed information about the strains see text). Cultures where grown in a micro-cultivation platform (*n* = 3 for each clone) in Delft media with 10 g/L glucose. For the growth curves (**a**–**d**) the means of the triplicates without the standard deviation are shown to ensure readability of the data. “Green value” corresponds to the biomass density. The stars indicate the clone selected to be further characterized in shake flasks and bioelectrochemical system (BES) experiments (where applicable).

**Figure 3 microorganisms-08-01959-f003:**
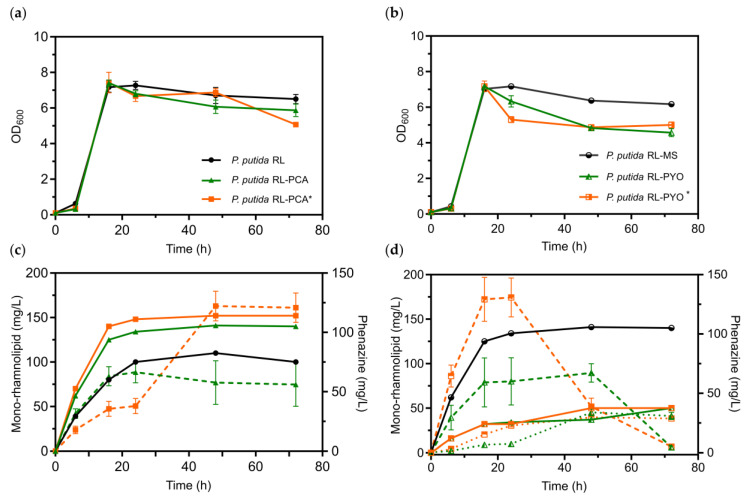
Fully aerobic flasks cultivation of different *P. putida* strains for mono-rhamnolipid and phenazine production (PCA and PYO). Non-phenazine producing control strains shown in black traces (circles). Phenazine production strains are shown in green (triangles). *: doubled amount of salicylate inducer (2 mM vs. normally 1 mM; orange squares). Growth measurement via optical density at 600 nm for (**a**) *P. putida* RL-PCA strain and *P. putida* RL control and (**b**) *P. putida* RL-PYO and *P. putida* RL control in Delft minimum media containing 10 g/L glucose. (**c**,**d**) show the respective rhamnolipid production (solid traces), PCA production (dashed traces) and PYO production (dotted traces) of these cultivations. (*n* = 3; error bars represent standard deviations).

**Figure 4 microorganisms-08-01959-f004:**
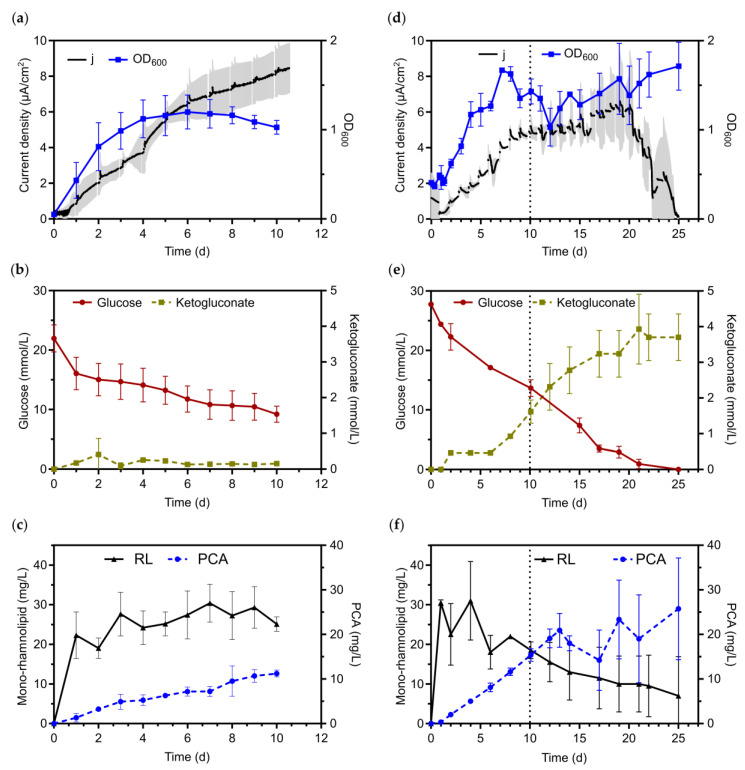
Passively aerated BES reactors of *P. putida* RL-PCA at an applied potential of 0.2 V. (**a**–**c**) Benchtop BES reactors (*n* = 3), showing data for cell density (OD_600_) and current production, glucose consumption and 2-ketogluconate production, formation of PCA and rhamnolipids (RL). (**d**–**f**) Electrobioreactors (*n* = 2), with the equivalent data shown as for the benchtop reactors. The dotted lines indicate the length of the benchtop BES reactor experiment. Gluconate and acetate were not detected in these experiments and are therefore not shown.

**Table 1 microorganisms-08-01959-t001:** Bacterial strains and plasmids used for tailoring heterologous rhamnolipid production with phenazine producing *P. putida*.

Strains/Plasmids	Characteristics	Source
Strains		
*P. aeruginosa* PAO1	Wildtype	DSMZ
*P. putida* 14phz2	PCA producer	[19]
*P. putida* RL	RL producer	this study
*P. putida* RL-PCA	PCA and RL producer	this study
*P. putida* RL-MS	PhzMS and RL producer	this study
*P. putida* RL-PYO	PCA, PYO, and RL producer	this study
Plasmids		
pJNN.*rhlAB*	*ori*RO1600 for *Pseudomonas* and *ori*CoIE1 for *E. coli*; gentamycin resistance-cassette, salicylate-inducible *nagR*/pNagAa promoter, *rhlA* and *rhlB* genes (*P. aeruginosa* PAO1)	this study
pJNN.*rhlAB.phzMS*	*ori*RO1600 for *Pseudomonas* and *ori*CoIE1 for *E. coli*; gentamycin resistance-cassette, salicylate-inducible *nagR*/pNagAa promoter, *phzM, phzS, rhlA* and *rhlB* genes (*P. aeruginosa* PAO1)	this study
pBNT.14*phz2*	ORI: ori/IHF for replication in *E. coli* and *Pseudomonas*; kanamycin resistance-cassette, salicylate-inducible *nagR*/pNagAa promoter, *phzA2-G2* genes (*P. aeruginosa* PA14)	[19]
pSK02	*ori*R6K-pir dependent origin of replication; *oriI* -origin of transfer; *riboJ* gene; kanamycin resistance-cassette, *rhlA* and *rhlB* genes (*P. aeruginosa* PAO1)	[16]

**Table 2 microorganisms-08-01959-t002:** Recombinant rhamnolipid production in batch experiments in this study compared to other *Pseudomonas* producers.

Organism	Medium	Substrate (conc. in g/L)	Duration	Maximal RL Titer (g/L)	Growth Condition	Ref.
*P. putida* KT42C1 pVLT31_*rhlAB*	LB	Glucose (10)	42 h	1.5	Flask, aerobic	[11]
*P. putida* KT2440 pSynPro8	LB	Glucose (10)	22 h	3.22	Flask, aerobic	[10]
*P. putida* KT2440 * pSynPro8oT	ModR	Glucose (253)	72 h	14.9	Benchtop bioreactor, aerobic	[38]
*P. putida* KT2440 Δ*flag* SK4	Delft	Glucose (11)	10 h	1.48	Benchtop bioreactor, aerobic	[16]
*P. putida* RL KT2440 pJNN.*rhlAB*	LB	Glucose (10)	30 h	0.73	Microscale, aerobic	this study
*P. putida* RL KT2440 pJNN.*rhlAB*	Delft	Glucose (10)	30 h 72 h	0.36 0.11	Microscale, aerobic Flask, aerobic	this study
*P. putida* RL-PCA KT2440 pJNN.*rhlAB*_pBNT.14*phz2*	Delft	Glucose (10)	30 h 72 h	0.11 0.15	Microscale, aerobic Flask, aerobic	this study
*P. putida* RL-PCA KT2440 pJNN.*rhl*AB_pBNT.14*phz2*	Delft	Glucose (5)	10 d	0.02	Benchtop BES, PA **	this study
*P. putida* RL-PCA KT2440 pJNN.*rhlAB*_ pBNT.14*phz2*	Delft	Glucose (5)	25 d	0.03	Electrobioreactor BES, PA **	this study

* Data in this row represent the highest titer achieved with recombinant *P. putida*. However, this process was operated in fed-batch with a continuous glucose feed and does not allow for direct comparison with other presented data from batch operations. ** PA: passive aeration via open vent filters.

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
