# Peer review of "Coupling an Electroactive Pseudomonas putida KT2440 with Bioelectrochemical Rhamnolipid Production"

_microorganisms, 2020, doi:10.3390/microorganisms8121959_

Round 1

Reviewer 1 Report

Review of the manuscript ID Microorganisms-1027221, entitled "Coupling an electroactive Pseudomonas putida KT2440 with bioelectrochemical rhamnolipid production"

Generalities

The manuscript entitled “Coupling an electroactive Pseudomonas putida KT2440 with bioelectrochemical rhamnolipid production” explores how rhamnolipids can be produced using Pseudomonas putida KT2440 in a BES. The article is interesting and worth to publish. I only have few comments.

Specifications

  • A tentative pathway for rhamnolipids productin would be appreciated.
  • Line 89. Passive aeration and electrobioreactors were tested at 27 mM glucose, while active aeration was tested at 55 mM glucose. Why?
  • Line 133. How were the Ag/AgCl assembled? A reference or a description in the supplementary material would be worthy for the audience.
  • Lines 134 and 154. I think that these sentences should be modified to clearly state that the anode was potentiostatically controlled at +0.2 V.
  • Line 151. According to this sentence, the reactor was equipped with a pO2 However, I could not find any value of oxygen concentration in the manuscript. It could be worthy to explain what “active aeration” or “passive aeration” means in terms of oxygen concentration.
  • Lines 218 and 236. The number of decimals should be the same for Mean and SD.
  • Lines 257-259. Authors could explain a little bit deeper the mechanism behind this process.
  • Line 293. Lower cell density because of the formation of a biofilm?
  • Discussion section. Results obtained indicates that the system needs to be operated with passive aeration. This aeration is obtained by opening the vent filters. It implies that the results oxygen distribution inside the reactor will depend on the reactor’s set up (Vent diameter, length and diameter of the reactor, etc.), and thus, different reactors could bring different results. If available, could the authors discuss about the oxygen concentrations detected when working at active or passive aeration? Which oxygen set-point is needed for the production of rhamnolipids?

Reviewer 2 Report

The presented manuscript concerns rhamnolipid along with phenazine and electrical current production. The study is well planned and performed and based on interdisciplinary examinations reasonable conclusions are made. I recommend publication. What I want to draw the authors' attention to is that they show only 2 figures in the manuscript, while in the additional material 8 S-figures are presented. Would it not be good to select and present some of the additional figures as main in the manuscript?
